# Electrochemical Biosensors Based on Carbon Nanomaterials for Diagnosis of Human Respiratory Diseases

**DOI:** 10.3390/bios13010012

**Published:** 2022-12-22

**Authors:** Chunmei Li, Bo Che, Linhong Deng

**Affiliations:** Changzhou Key Laboratory of Respiratory Medical Engineering, Institute of Biomedical Engineering and Health Sciences, School of Medical and Health Engineering, Changzhou University, Changzhou 213164, China

**Keywords:** electrochemical biosensors, respiratory diseases, carbon nanomaterials

## Abstract

In recent years, respiratory diseases have increasingly become a global concern, largely due to the outbreak of Coronavirus Disease 2019 (COVID-19). This inevitably causes great attention to be given to the development of highly efficient and minimal or non-invasive methods for the diagnosis of respiratory diseases. And electrochemical biosensors based on carbon nanomaterials show great potential in fulfilling the requirement, not only because of the superior performance of electrochemical analysis, but also given the excellent properties of the carbon nanomaterials. In this paper, we review the most recent advances in research, development and applications of electrochemical biosensors based on the use of carbon nanomaterials for diagnosis of human respiratory diseases in the last 10 years. We first briefly introduce the characteristics of several common human respiratory diseases, including influenza, COVID-19, pulmonary fibrosis, tuberculosis and lung cancer. Then, we describe the working principles and fabrication of various electrochemical biosensors based on carbon nanomaterials used for diagnosis of these respiratory diseases. Finally, we summarize the advantages, challenges, and future perspectives for the currently available electrochemical biosensors based on carbon nanomaterials for detecting human respiratory diseases.

## 1. Introduction

Respiratory diseases affect the nasal cavity, bronchi, lungs, chest and other parts of the human body, and can be either non-infectious, as in asthma, chronic obstructive pulmonary disease (COPD), chronic bronchitis and idiopathic pulmonary fibrosis, or infectious, as in pulmonary infections caused by viruses, bacteria and other microorganisms [1]. With the increasing extent of air pollution, smoking, aging population and other factors, respiratory diseases will also have an increasingly serious impact on the health of the people, as a result of increased incidence rate and mortality [2]. At present, respiratory diseases can be diagnosed with a variety of methods. For example, pulmonary fibrosis can be diagnosed by high-resolution computed tomography (HRCT) with high sensitivity [3]. And coronavirus disease 2019 (COVID-19) can be diagnosed by detection of SARS coronavirus 2 (SARS-CoV-2) using heterogeneous serological methods in the laboratory [4]. However, these methods are either time-consuming, expensive, or require further diagnostic examinations such as surgical lung biopsy and a multiple disciplinary consultation for diagnosis [5]. Therefore, it is urgent to develop simple, fast-operating diagnostic procedures for diagnosis of respiratory diseases in the early stage, such as those assisted by detection of respiratory viruses, related DNA fragments, proteins, or RNA with electrochemical biosensors. 

The electrochemical biosensor is an important branch of the electrochemical sensor, which uses electrode as energy exchange element [6]. Compared with other types of sensors, the electrochemical biosensor is based on direct electronic signals, such as ampere, volt ampere and impedance changes [7,8]. The transduction process of electrochemical biosensors can be completed in a short space of time in the electrochemical workstation, greatly reducing the test’s time and cost [9,10,11,12]. In addition, the high sensitivity of electrochemical biosensors can be guaranteed by using biometric components with high specificity and affinity, or modifying electrodes with unique materials with distinctive electrical properties [9,13,14,15]. Electrochemical biosensors have been widely used in the field of analysis with their unique advantages, and have been especially used to detect respiratory viruses, related DNA fragments, proteins or RNA, to assist the diagnosis of respiratory diseases over recent years [16,17,18,19,20,21,22,23,24]. 

Carbon nanomaterials, less than 100 nm in size in at least one dimension, are composed of carbon atoms and non-carbon atoms, in which carbon atoms are commonly sp^2^ and sp^3^ hybridization [25]. These materials come in different forms including carbon quantum dots, graphene quantum dots, carbon nanotubes, graphene, graphitic carbon nitride (also known as g-C_3_N_4_), fullerene and diamond [26]. They all have excellent physical, chemical, mechanical and electrical properties. In addition to good biocompatibility and bioactivity, carbon nanomaterials can be explored in one way or the other to fabricate biosensors [27]. Such biosensors have also shown great potential for diagnosis of respiratory diseases. For example, a biosensor fabricated with carbon nanotubes has been developed for diagnosis of lung cancer via the detection of long non-coding RNAs (lncRNAs), specifically, the biomarker metastasis-associated lung adenocarcinoma transcript 1 (MALAT1) in the blood [28]. And biosensors fabricated with carbon nanodots and graphene quantum dots have been demonstrated to be capable of detecting the presence of SARS-CoV-2, and Haemophilus influenza virus by monitoring of Haemophilus influenza genome in human plasma samples, respectively [29,30].

In this review, we will provide an overview of the most recent developments of electrochemical biosensors fabricated with carbon nanomaterials and their composites in the field of diagnosis of respiratory diseases (Figure 1). We will focus on several common respiratory diseases, including influenza, pulmonary fibrosis, tuberculosis, lung cancer and COVID-19, in terms of their characteristics and diagnosis by using electrochemical biosensors based on carbon nanomaterials. Last but not least, we will briefly discuss the challenges and future perspectives of electrochemical biosensors based on carbon nanomaterials for the diagnosis of respiratory diseases. 

## 2. Characteristics of Carbon Nanomaterials and Human Common Respiratory Diseases

Carbon nanomaterials are materials less than 100 nm in size at least in one dimension. They are composed of carbon atoms and non-carbon atoms, in which carbon atoms are commonly sp^2^ and sp^3^ hybridization. Generally, they include carbon quantum dots, graphene quantum dots, carbon nanotubes, graphene, graphitic carbon nitride (also known as g-C_3_N_4_), fullerene and diamond, and they all have excellent physical, chemical, mechanical and electrical properties. For more information related to carbon nanomaterials, please read a review I wrote earlier [25]. 

The respiratory system consists of the respiratory tract and lungs, wherein the respiratory tract consists of the nose, throat, larynx, trachea, bronchus and various bronchial branches in the lungs [31]. Respiratory diseases refer to the diseases in which lesions are located in the respiratory system. Common respiratory diseases include the infectious influenza, acute tracheobronchitis, chronic bronchitis, tuberculosis and COVID-19, and the non-infectious asthma, COPD, pulmonary fibrosis, lung cancer and so on [32,33,34]. Here we focus on influenza, COVID-19, pulmonary fibrosis, tuberculosis and lung cancer.

Influenza is an infectious acute respiratory disease caused by the influenza virus. Its clinical characteristics are acute infection, obvious symptoms, such as high fever, headache, systemic pain, weakness, etc. Influenza is mainly transmitted through contact and droplets [35]. It is highly infectious. There are various types of influenza viruses, including swine influenza virus, avian influenza virus, and influenza A, B, and C [36,37,38,39,40,41].

COVID-19 is another contagious respiratory disease, brought on by SARS-CoV-2, and first identified in 2019 [42]. SARS-CoV-2 is polymorphic or usually spherical, with a diameter range of 80–160 nm, and contains a single-positive strand RNA genome of about 30 kb with a 5′ cap structure and a 3′ poly(A) [43]. The 3′ poly(A) tail of SARS-CoV-2 RNA genome can encode four main structural proteins, namely, spike (S) protein, envelope (E) protein, membrane (M) protein and nucleocapsid (N) protein [44]. When the S protein of SARS-CoV-2 binds to a person’s cell’s surface receptor, angiotensin converting enzyme 2 (ACE2), he will be infected [43,44]. SARS-CoV-2 transmits primarily through respiratory droplets by inhalation of sneezing, coughing, talking and exhaled droplets of gas. People who are infected may have typical symptoms, such as coughs or sneezes, or may not have symptoms [45]. In the process of transmission, the single-positive strand RNA genome of SARS-CoV-2 replicates continuously over time, and a variety of variants will appear. Currently, there are several SARS-CoV-2 variants, such as the Alpha, Beta, Gamma, Delta and Omicron variants [46,47]. Since December 2019, COVID-19 has greatly affected our lives and led to an unprecedented socio-economic burden.

Pulmonary fibrosis is a severe and long-lasting interstitial respiratory disease brought on by aggregation of fibroblasts and deposition of lung extracellular matrix, and more serious pulmonary fibrosis is commonly accompanied by malignant reaction and damage to the lung cells/tissue structure [48]. According to pathogenic factors, clinical presentation, relative responsiveness to immunosuppression and imaging characteristics, pulmonary fibrosis can be divided into primary pulmonary fibrosis, secondary pulmonary fibrosis, idiopathic pulmonary fibrosis, pulmonary interstitial fibrosis, interstitial pneumonia and pulmonary fibrosis caused by drugs or emission lines, and so on [20]. In recent years, pulmonary fibrosis, especially idiopathic pulmonary fibrosis, has affected millions of people and its incidence rate is rising year by year.

Tuberculosis is a chronic respiratory disease caused by a bacterium called *Mycobacterium tuberculosis* attacking the lungs [16,17,18]. In the initial stage, people with latently infected tuberculosis may only exhibit minor symptoms or none at all, and most early infections can only be found by X-ray examination [49]. When tuberculosis disease is serious, patients infected by *Mycobacterium tuberculosis* are hemoptysis, and then tuberculosis disease can be fatal if not treated properly [49]. Lung cancer is a disease in which lung cells proliferate out of control [50]. There are two main sub-type lung cancers, namely small cell lung cancer (including oat cell carcinoma, intermediate cell carcinoma and compound oat cell carcinoma) or non-small cell lung cancer (including adenocarcinoma and squamous cell carcinoma), and the latter is more common than the former [51]. Lung cancer involves rapid proliferation and early extensive metastasis [22,23,52]. As a result, it may spread from one organ to another. For example, lung cancer may spread to lymph nodes or the brain. In comparison, cancer starting from other organs may also spread to the lungs [50]. The initial typical manifestation of lung cancer is cough and dyspnea caused by enlarged hilar mass and huge mediastinal lymph nodes, and it is more sensitive to radiotherapy and chemotherapy [50].

## 3. Electrochemical Biosensors Based on Carbon Nanomaterials for Diagnosis of Human Respiratory Diseases

Respiratory diseases often lead to dyspnea and shortness of breath in patients, which in turn leads to low blood oxygen saturation and gradual tissue hypoxia that can ultimately lead to coma and even death [31]. Once a human respiratory disease epidemic breaks out, there will inevitably be a shortage of medical personnel and equipment. At present, academic research efforts have focused on the treatment of acute large-scale epidemic respiratory diseases such as COVID-19 [53]. But the development of advanced routine diagnostic methods for respiratory diseases is still of great significance for the demand for early accurate detection of the diseases.

Currently widely-used routine diagnostic methods such as chest X-ray, polymerase chain reaction (PCR) detection, Xpert MTB/RIF and immunological detection [54,55] are defective in speed for analysis, sensitivity, discriminatory power and specificity. In contrast, electrochemical biosensors are known to be advantageous in simplicity, speed, sensitivity, and low operation cost, which makes them potential alternative tools for routine diagnosis of human respiratory diseases and recently has received great attention from the research community [6,17,18,19,37,56].

Furthermore, carbon nanomaterials have proved to be superior for fabrication of electrochemical biosensors. By definition, carbon nanomaterials are carbon-based materials composed of sp^2^ and sp^3^-bonded carbon atoms or heterogeneous components (non-carbon atoms) with at least one dimension of less than 100 nm [25]. These materials come in different forms, including carbon quantum dots, graphene quantum dots, carbon nanotubes, graphene, graphitic carbon nitride (also known as g-C_3_N_4_), fullerene and diamond. But they all have exceptional physicochemical and biological qualities. For example, carbon quantum dots or graphene quantum dots are well-known for their small diameters (∼5 nm), chemical inertness, minimal toxicity, superior biocompatibility and photoluminescent stability [57,58]. On the other hand, carbon nanotubes can provide a relatively large surface area with unique physical and chemical properties and surface functionalization ability. Graphene or graphene oxide is known for its superior water solubility, in addition to a huge surface area with distinctive surface properties and minimal cytotoxicity [59]. Further, g-C_3_N_4_ (bulk, nanosheets, and quantum dots) possesses exceptional optical qualities, chemical and thermal stability together with biocompatibility and minimal toxicity [60]. Fullerene is a common zero-dimensional carbon material associated with large conductivity and strong electron receptivity as well as unique redox activity [20,61]. And in addition to good biocompatibility and bioactivity, diamond has been reported to have excellent fluorescent capacity, and a cost-effective advantage for large-scale manufacturing of medical devices [62,63]. These superior properties of carbon nanomaterials can be explored in one way or the other to fabricate tailor-made electrochemical biosensors for diagnosis of various human respiratory diseases.

Currently, a series of electrochemical biosensors based on carbon nanomaterials have been developed and widely used for the diagnosis of respiratory diseases, demonstrating their great potential in clinical applications. For example, as COVID-19 broke out in 2019, electrochemical biosensors fabricated with graphene oxide that had been functionalized to target specific RNA of SARS-CoV-2 were used for the diagnosis of COVID-19 [64]. For diagnosis of lung cancer, electrochemical biosensors were fabricated with three-dimensional (3D) graphene to identify two lung cancer biomarkers, cytokeratin 19 fragment 21-1 and carcinoembryonic antigen [65]. In addition, multi-walled carbon nanotubes have been used to develop an electrochemical DNA biosensor to detect *Mycobacterium tuberculosis* bacteria for the diagnosis of tuberculosis. More details of these electrochemical biosensors based on carbon nanomaterials will be described and discussed respectively in the following.

### 3.1. Electrochemical Biosensors for Influenza Diagnosis

Influenza viruses, including influenza A, B, and C viruses as well as the swine and avian influenza viruses., have become an increasingly serious hazard to human health. Every year, especially in the winter, a large number of people are infected by influenza virus via its directly crossing the human immune barriers. Obviously, effective methods for detecting the presence of the highly infectious influenza virus are urgently needed for monitoring and controlling the spread of viral infection among the population. Electrochemical methods have proven to be excellent alternative options for detecting influenza virus, with antibodies or nucleic acid as recognition reagents [66,67,68]. Recently, carbon nanomaterials have been used to fabricate electrochemical biosensors for detection of Haemophilus influenza, influenza A virus (such as H1N1, H5N1 and H7N9 influenza A virus), avian influenza virus (such as H5N1, H7N9, and H9N2) and swine influenza virus (such as H1N1 swine influenza virus) [30,36,41,69,70].

Specifically, Anik et al. have developed an influenza A biosensor that used an Au-screen printed electrode modified with graphene gold hybrid nanocomposite for an electrochemical impedance spectroscopy (EIS) analysis and demonstrated these biosensors for successful detection of the real influenza virus A (H9N2), as shown in Figure 1a [71]. Reddy et al. have developed influenza biosensors using a nickel oxide (NiO)-reduced graphene oxide (rGO)/MXene nanocomposite for detection of both active influenza viruses (H1N1 and H5N2) and influenza proteins via electrochemical signals (Figure 1b) [70].

Kinnamon et al. have fabricated a textile screen-printed influenza electrochemical biosensor for detecting influenza A virus exposed to the environment, using graphene oxide as transduction film of the textile screen-printed electrode (Figure 1c) [39]. Interestingly, Liu et al. have developed an electrochemical biosensor for detecting H5N1 gene sequence of avian influenza virus (AIV), in which multi-wall carbon nanotubes (MWNT), together with polypyrrole nanowires (PPNWs) and gold nanoparticles (GNPs), was used to fabricate a hybrid nanomaterial-modified electrode for immobilized DNA aptamer (Figure 1d) [37].

These electrochemical biosensors based on carbon nanomaterials for diagnosis of influenza described above have excellent performance, and details are shown in Table 1.

### 3.2. Electrochemical Biosensors for COVID-19 Diagnosis

The pandemic of COVID-19 brought on by SARS CoV-2 may become more serious under evolutionary pressure due to the emergence of transmissibility, pathogenicity and pathogenicity or SARS CoV-2 variants (such as Alpha, Beta, Gamma, Delta variant or Omicron variant) [72]. Additionally, SARS CoV-2 may become more adaptable and develop into a runaway form due to antibodies in COVID-19 convalescence or vaccine recipients. Traditional SARS CoV-2 detection methods mostly rely on laboratory technology, specifically, from initial virus culture, morphological observation and serological test to subsequent reverse transcription PCR, isothermal amplification technology, immunochromatography, and enzyme-linked immunosorbent immunofluorescence assay [73,74]. In comparison, the electrochemical biosensor is faster, more sensitive and accurate to identify and quantify SARS CoV-2, and therefore has become one of the most rapidly developing area in the field.

So far, it has been reported that a variety of COVID-19 electrochemical biosensors have been based on carbon nanomaterials [75]. The carbon nanomaterials used to fabricate electrochemical biosensors include, but are not limited to, graphene, graphene oxide nanocolloids, boron-doped diamond and functionalized graphene oxide [42,47,76]. And it has also been demonstrated that the electrochemical biosensors based on carbon nanomaterials could be used to successfully detect a DNA sequence corresponding to SARS-CoV-2, a SARS-CoV-2 nucleocapsid protein, a RNA of SARS-CoV-2, a protein sequence of the N protein of SARS-CoV-2, a SARS-CoV-2 spike protein, a SARS-CoV-2 S1 antigen, and SARS-CoV-2 variant Delta and a SARS-CoV-2 S protein, and so on [29,62,64], [77,78].

Figure 2 and Figure 3 show the design of some of these biosensors and the procedure for detecting SARS-CoV-2. In the first case, Ramanathan et al. have exploited a portable electrochemical biosensor for detecting SARS-CoV-2 nucleocapsid protein (NCP) as diagnosis of COVID-19, which used a gap-sized gold interdigitated electrode (AuIDE) deposited with ~20 nm diamond (Figure 2a) [62]. In the second case, Beduk et al. have developed a point-of-care (POC) COVID-19 diagnostic, in which a laser-scribed graphene (LSG)-based biosensing platform was built based on a miniaturized electrochemical sensing scheme combined with 3D gold nanostructures (Figure 2b) [46]. In the third case, Zhao et al. developed a portable electrochemical smartphone system for remote diagnosis of COVID-19, in which an electrochemical biosensor was fabricated with calixarene functionalized graphene oxide based on a super-sandwich-type recognition strategy, wherein calixarene functionalized graphene oxide was used to target RNA of SARS-CoV-2. The electrochemical biosensor has been confirmed to effectively detect the RNA of SARS-CoV-2 in the absence PCR and reverse-transcription process (Figure 3a) [64]. In the fourth case, Zamzami et al. have developed an electrochemical biosensor based on carbon nanotube field-effect transistor (CNT-FET) for detecting a SARS-CoV-2 S1 antigens in saliva samples, and this detection has been shown to be fast (2–3 min), quantitative, easy to use, and at a low cost (Figure 3b) [79].

These electrochemical biosensors based on carbon nanomaterials for diagnosis of COVID-19 described above have excellent performance; the details are shown in Table 2.

### 3.3. Electrochemical Biosensors for Pulmonary Fibrosis Diagnosis 

Pulmonary fibrosis is a severe chronic and progressive interstitial respiratory disease, and it has typical clinical symptoms such as dyspnea and dry cough. At present, high-resolution computed tomography (HRCT) is extensively used for the screening diagnosis of pulmonary fibrosis, which is highly sensitive. However, some patients do not show typical HRCT features and require further diagnostic examinations with surgical lung biopsy. The procedure of surgical lung biopsy is invasive and causes great pain to the patient. Therefore, it is desirable to develop alternative non-/minimal invasive procedures to assist in the diagnosis of pulmonary fibrosis. Electrochemical biosensors based on carbon nanomaterials have appeared to meet this requirement by detecting biomarkers of pulmonary fibrosis. For example, Zuo et al., have proposed an electrochemical biosensor using fullerene (C60) as electrode materials for detecting miR-3675-3p in human serum, which is known as a promising biomarker for pulmonary fibrosis [20]. Electrochemical biosensors based on a carbon-nanodots-modified screen-printed gold electrode as a transducer for gene detection, or based on carbon nanofibers for protein detection of cystic fibrosis transmembrane regulator (CFTR) as a biomarker of pulmonary fibrosis, have been reported, respectively [80,81]. Bonanni et al., have reported an electrochemical biosensor for diagnosis of pulmonary fibrosis, one based on gold nanoparticles in a graphite-epoxy nanocomposite (nanoAu-GEC) for the detection of triple base mutation deletion in a human DNA sequence related cystic-fibrosis [48].

These electrochemical biosensors based on carbon nanomaterials for diagnosis of pulmonary fibrosis described above have excellent performance, and the details are shown in Table 3.

### 3.4. Electrochemical Biosensors for Tuberculosis Diagnosis

Tuberculosis is a global public health concern, as one of the top 10 causes of death in the world. Therefore, it is crucial to have high-quality diagnosis of the disease. Conventional methods of diagnosis include culture-based/culture-independent methods, imaging-based methods, antigen detection, serological tests, PCR assay and so on, which all aim to identify the presence of *Mycobacterium tuberculosis* directly or indirectly [82,83]. But these conventional methods are known for limitations in terms of sensitivity, specificity, delayed response time, need for skilled personnel and expensive instrumentation [84,85,86,87]. In comparison, tuberculosis detection methods based on electrochemical biosensors would have the advantages of cost effectiveness, detection speed and accuracy, as well as excellent biological and chemical properties if the biosensors are fabricated with carbon nanomaterials. Therefore, a variety of electrochemical biosensors have been recently reported for the detection of *Mycobacterium tuberculosis* or its biomarkers [84,88,89]. These biosensors have mainly been fabricated with carbon nanomaterials, including graphene oxide nanoribbons [16], carbon nanotubes [18], fullerene nanoparticles [19], graphene oxide [90], 3D graphene [91], graphene quantum dot [92], and nitrogen-doped carbon nanodots [93]. For tuberculosis detection, these biosensors generally detected *Mycobacterium tuberculosis* [90], while some detected biomarkers of *Mycobacterium tuberculosis* such as CFP10-ESAT6 antigen and ESAT-6 antigen [86,92,94], *Mycobacterium tuberculosis* DNA sequence [17,18,91], interferon gamma (IFN-γ) [95] and methyl nicotinate (metabolite of *Mycobacterium tuberculosis*) [96] and so on.

Figure 4 and Figure 5 further illustrate the design of some of these electrochemical biosensors based on carbon nanomaterials and the procedures used for detecting *Mycobacterium tuberculosis* or its biomarkers. The first is an electrochemiluminescence (ECL) biosensor fabricated with self-enhanced ruthenium (Ru) II-based nanocomposite (NCNDs-BPEI-Ru) which has proved to be ultrasensitive for detection of *Mycobacterium tuberculosis* (Figure 4a) [93]. The NCNDs-BPEI-Ru nanocomposite was synthesized using nitrogen-doped carbon nanodots (NCNDs), tris (4,4′-dicarboxylicacid-2,2′-bipyridyl) Ru II dichloride (Ru(dcbpy)_3_Cl_2_), polyethyleneimine (BPEI). The second is another ECL biosensor fabricated with gold nanoparticle-coated magnetic beads (AuNP@MB), and the AuNP@MB was attached on a nanofiber prepared using graphene oxide and polyaniline, namely GO-PANI-NF. The ECL biosensor proved to be a very sensitive, real-time and dynamic sensor for detecting IFN-γ in blood as a biomarker for latent infection of *Mycobacterium tuberculosis* (Figure 4b) [95]. The third is a universal amperometric DNA biosensor fabricated with carbon nanotubes doped with polyaniline (CNTs-PAN) nanohybrid. The CNTs-PAN nanohybrid was a flower-like structure which can provide a large surface area with abundant active groups and efficient redox activity to form a tracer label. The flower-like CNTs-PAN nanohybrid can generate and amplify the electrochemical signal, resulting in ultra-sensitive detection of the specific IS6110 DNA sequence of *Mycobacterium tuberculosis* (Figure 5a) [18]. The last is an electrochemical biosensor fabricated with gold nanoparticles (AuNPs) immobilized over reduced graphene oxide nanoribbons (RGONRs), which was developed for detecting target *Mycobacterium tuberculosis* (Figure 5b) [16].

These electrochemical biosensors based on carbon nanomaterials for diagnosis of tuberculosis described above have excellent performance, and details are shown in Table 4.

### 3.5. Electrochemical Biosensors for Lung Cancer Diagnosis

Lung cancer has been a major concern worldwide due to the highest morbidity and mortality rate associated with the disease. Therefore, the development of adequate techniques for detecting lung cancer biomarkers is urgently required for close monitoring of the patients. And several electrochemical biosensors based on carbon nanomaterials have shown great potential in this regard.

As shown in Figure 6a, Zhuo et al., have developed a novel immuno-electrochemical biosensor for sensitive detecting a specific small cell lung cancer (SCLC) biomarker, progastrin releasing-peptide (ProGRP). The electrode of the biosensor was fabricated with Au nanoparticle/graphene together with ferrocene and glucose oxidase-multifunctional Au/TiO_2_ nanocomposites, in which Au nanoparticle/graphene served as an antibody immobilization matrix [52]. On the other hand, Chen et al. have developed a transistor-based electrochemical biosensor for sensitively and conveniently detecting sialic acid level in serum samples. Sialic acid residues are generally highly expressed by cancer cells, which can be used as lung cancer biomarker. The transistor-based electrochemical biosensor consisted of three standard electrodes. One of the electrodes was modified with carboxylated multi-wall carbon nanotubes, which can produce the drain-source channel current signal; therefore, sialic acid in serum samples can be sensitively detected (Figure 6b) [97].

Furthermore, Jafari-Kashi et al. have designed a label-free electrochemical DNA-biosensor for the early diagnosis of lung cancer via the detection of lung cancer biomarker, cytokeratin 19 fragment 21-1 (CYFRA21-1). The biosensor electrode was modified with reduced-graphene oxide, poly pyrrole, silver nanoparticles and single-strand DNA for capture of the lung cancer biomarker (Figure 7a) [24]. Similarly, Choudhary et al. used carbon nanotubes and chitosan (CNT-CHI) composite to develop a label-free electrochemical immunosensor for simultaneously detecting anti-MAGE A2 and anti-MAGE A11, which are also known biomarkers of lung cancer (Figure 7b) [51].

These electrochemical biosensors based on carbon nanomaterials for diagnosis of lung cancer described above have excellent performance; details are shown in Table 5.

### 3.6. Electrochemical Biosensors for Other Human Respiratory Diseases Diagnosis

In addition to the above electrochemical biosensors based on carbon nanomaterials for diagnosis of influenza, COVID-19, pulmonary fibrosis, tuberculosis and lung cancer, there are other reports about electrochemical biosensors based on carbon nanomaterials for diagnosis of other human respiratory diseases such as allergic rhinitis, Middle East respiratory syndrome (MERS), deep vein thrombosis, asthma and pneumonia [98,99,100,101]. These electrochemical biosensors have been fabricated by using carbon nanomaterials such as graphene oxide, reduced graphene oxide and carbon nanotubes for detection biomarkers/microorganism of human respiratory diseases; the biomarkers include tryptase, MERS nanovesicle, D-dimer and pathogenic microorganisms. Additionally, these electrochemical biosensors also have excellent performance; details are shown in Table 6.

## 4. Comparative Analysis with Conventional Approaches for the Diagnosis of Human Respiratory Diseases

In this review, we summarize electrochemical biosensors based on carbon nanomaterials for diagnosis of human respiratory diseases. Compared with other biosensors for diagnosis of human respiratory diseases, the biosensors summarized in this review not only show the advantages of carbon nanomaterials, such as stable properties and easy preparation, but also show the advantages of electrochemical strategies such as low detection limit and fast reaction time. Taking COVID-19 as an example, Table 7 shows details of comparisons of different biosensors for diagnosis of human respiratory diseases with electrochemical biosensors based on carbon nanomaterials. 

## 5. Conclusions and Outlooks

In summary, this article provides a concise overview of recent studies on electrochemical biosensors based on carbon nanomaterials for the diagnosis of several important human respiratory diseases, including influenza, COVID-19, pulmonary fibrosis, tuberculosis and lung cancer. We firstly summarize the electrochemical application of carbon nanomaterials for diagnosis of various respiratory diseases in one review. Compared with other reviews related to human respiratory diseases, this review provides a comprehensive analysis of the unique characteristics of carbon nanomaterials and the advantages of electrochemical assays for diagnosis of human respiratory diseases. Besides, compared with other traditional methods for diagnosis of human respiratory diseases, these electrochemical biosensors based on carbon nanomaterials which are summarized in this review are simple, fast-operating diagnostic procedures for diagnosis of respiratory diseases in the early stage by detection of respiratory viruses, related DNA fragments, proteins or RNA. In addition, these biosensors have shown great potential for both basic research and clinical applications in the field of alerting and preventing the spread of respiratory diseases. 

There are, however, still many significant challenges that need to be overcome before the research and development of these biosensors successfully translate into clinical practices. The challenges include, but are not limited to (1) the fact that the current electrochemical biosensors based on carbon nanomaterials can detect only a limited number of respiratory viruses, related DNA fragments, proteins and RNAs, (2) carbon nanomaterials as electrode materials in some electrochemical biosensors for the diagnosis of respiratory diseases only play a small role and their excellent properties have not been fully utilized, which is far from the materials’ full potential, and (3) the current range of electrochemical biosensors based on carbon nanomaterials have some restrictions, such as restrictions in terms of the limit of detection and range of linearity, in the diagnosis of respiratory diseases.

Thus, in future research and development of electrochemical biosensors based on carbon nanomaterials for diagnosis of respiratory diseases, efforts should be made to exploit not only more biomarkers related to respiratory diseases, but also more functions of the carbon nanomaterials so to develop them into mainstream materials for fabrication of electrochemical biosensors. Further, the detection limit and range of linearity of the electrochemical biosensor based on carbon nanomaterials for diagnosis of respiratory diseases should be optimized. In addition, new synthetic methods of carbon nanomaterials should be exploited to make them possess better optical, electrical and other properties. Finally, due to the excellent application potential of carbon nanomaterials in electrochemical biosensors for diagnosis of human respiratory diseases, carbon nanomaterials should be exploited to be the main electrode materials. 

## Data Availability

Not applicable.

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
