# Peer review of "Electrochemical Biosensors Based on Carbon Nanomaterials for Diagnosis of Human Respiratory Diseases"

_biosensors, 2022, doi:10.3390/bios13010012_

Round 1

Reviewer 1 Report

In this work, the Authors exposed a background of the "Electrochemical Biosensors Based on Carbon Materials for Diagnosis of Human Respiratory Diseases"; however I consider that the article requires important changes in its structure mainly by:

1. Important problems with the language. In some ways, it is a bit informal as well as important mistakes in the grammar and the format of the references.

2. I consider that the authors have focused the work mainly on brief and wide concepts of respiratory illness instead of the analysis of the different sensor platforms. Additionally, the conceptualization and the main advantages of the carbon materials are missing during the revision.

3. In some paragraphs, there are some "ideas" that sound like statements (i.e. Line 97) in which a reference is required.

4. In terms of the COVID, please check the following IMPORTANT references:

-Can graphene take part of the in the fight against COVID-19?

-Rapid Detection of COVID-19 Causative Virus (SARS-CoV-2) in Human Nasopharyngeal Swab Specimens Using Field-Effect Transistor-Based Biosensor

-Prospects of nanomaterials-enabled biosensors for COVID-19 detection

and the list of references can be longer. Please check the bibliography properly.

5. I consider that the state-of-art also must consider a deep analysis of the mechanism of detection, the type of platform, and the most important figures of merit such as LOD, LOQ, Lineal range, real sample analysis, technique, sensitivity, speed of the measurement. I think that this kind of revision, also, must include these parameters to make it more interesting to the reader.

Reviewer 2 Report

Please check the file

Reviewer 3 Report

In this review article, authors briefly introduce the working principle and fabrication of various electrochemical biosensors based on carbon nanomaterials for diagnosis of these respiratory diseases. It is by no doubt an interesting topic.

Overall, I found the review to be comprehensive but too qualitative and even vague in some sections. The authors should include much more specific and quantitative information when presenting the results of the studies they are discussing. This is particularly important when comparing these novel detection strategies to more conventional approaches, specifically with regard to sensitivity, accuracy, precision and overall performance. I think there needs to be more critical analysis throughout the paper. The authors have some text on this but parts of the paper still read like a laundry list of studies.

The Conclusion and outlooks part are suggested to give further advice and review for future research. 

Round 2

Reviewer 1 Report

Thanks to the author to consider the comments and suggestions. The manuscript is acceptable now for publication.

Author Response

Thank you again for your great contribution to the improvement of our manuscript

Reviewer 3 Report

It could be accepted in present form. 

Author Response

(The authors gave the same response as above.)
